# The Neuroprotective Effects of Melatonin: Possible Role in the Pathophysiology of Neuropsychiatric Disease

**DOI:** 10.3390/brainsci9100285

**Published:** 2019-10-21

**Authors:** Jung Goo Lee, Young Sup Woo, Sung Woo Park, Dae-Hyun Seog, Mi Kyoung Seo, Won-Myong Bahk

**Affiliations:** 1Department of Psychiatry, College of Medicine, Haeundae Paik Hospital, Inje University, Busan 47392, Korea; iybihwc@inje.ac.kr; 2Paik Institute for Clinical Research, Department of Health Science and Technology, Graduate School, Inje University, Busan 47392, Korea; swpark@inje.ac.kr; 3Department of Psychiatry, College of Medicine, The Catholic University of Korea, Seoul 07345, Korea; youngwoo@catholic.ac.kr; 4Department of Convergence Biomedical Science, College of Medicine, Inje University, Busan 47392, Korea; 5Department of Biochemistry, College of Medicine, Inje University, Busan 47392, Korea; daehyun@inje.ac.kr; 6Paik Institute for Clinical Research, Inje University, Busan 47392, Korea; banaba66@inje.ac.kr

**Keywords:** melatonin, oxydative stress, antioxidant, neuroprotective effect, antidepressant

## Abstract

Melatonin is a hormone that is secreted by the pineal gland. To date, melatonin is known to regulate the sleep cycle by controlling the circadian rhythm. However, recent advances in neuroscience and molecular biology have led to the discovery of new actions and effects of melatonin. In recent studies, melatonin was shown to have antioxidant activity and, possibly, to affect the development of Alzheimer’s disease (AD). In addition, melatonin has neuroprotective effects and affects neuroplasticity, thus indicating potential antidepressant properties. In the present review, the new functions of melatonin are summarized and a therapeutic target for the development of new drugs based on the mechanism of action of melatonin is proposed.

## 1. Introduction

Melatonin is a ubiquitous natural neurotransmitter-like compound (Figure 1) secreted by the pineal gland in the brain [1]. Melatonin has diverse functions that regulate the circadian rhythm, energy metabolism, and the immune system; it also inhibits oxidative stress and participates in the aging process [2,3]. In vertebrates, melatonin is produced in some tissues; however, most melatonin is produced by the pineal gland, secreted into the blood, and acts as a hormone [4]. In mammals, the pineal gland is a neuroendocrine gland located near the thalamus, and consists of pinealocytes that produce melatonin [4]. Melatonin is a hormone that regulates the circadian rhythm, and has been investigated in many studies [5]. However, in recent studies, melatonin has been shown to exert protective effects on neurons, especially against oxidative stress, and can be used as a target for the development of various neurological diseases [6,7,8,9,10]. In this review, the neuroprotective effects of melatonin that have recently attracted much interest, starting from the basic biology of melatonin, are reviewed and evaluated.

## 2. Synthesis, Metabolism and Action Mechanism of Melatonin

### 2.1. Synthesis and Metabolism of Melatonin

Melatonin is synthesized in the pinealocytes of the pineal gland, and its synthesis is regulated by the hypothalamic paraventricular nuclei [11]. In particular, hypothalamic paraventricular nuclei project to preganglionic sympathetic neurons of the spinal cord, and postganglionary sympathetic neurons of the superior cervical ganglia project to the pineal glands [12]. The sympathetic nerve terminal norepinephrine binds to the alpha and beta noradrenergic receptors located on the cell membrane of the pinealocytes, activating the *cAMP-PKA-CREB* and *PLC-Ca2+-PKC* pathways to initiate melatonin synthesis [13]. The synthesis of melatonin begins with the conversion of tryptophan to 5-hydroxytryptophan by tryptophan hydroxylase. Then, 5-hydroxytryptophan is converted to serotonin, which is converted to N-acetylserotonin by arylalkylamine N-acetyltransferase (*AANAT*) and, finally, to melatonin by acetylserotonin O-methyltransferase [14,15]. *AANAT* and hydroxyindole-O-methyltransferase (*HIOMT*) are important enzymes involved in the production of melatonin in the central nervous system (CNS). *AANAT* converts serotonin to N-acetylserotoin, which in turn converts N-acetylserotoin to melatonin [3]. In particular, *AANAT* is not considered a rate-limiting enzyme that regulates the synthesis of melatonin at night [16].

The hypothalamic suprachiasmatic nuclei play a central role in regulating melatonin secretion, consistent with the light/dark cycle of the day, leading to the release of melatonin, which is secreted more at night [17]. In contrast, light stimuli inhibit the synthesis of melatonin by promoting degradation of melanopsin in the retinal photoreceptive ganglion cell projected into the hypothalamus [17]. The pineal gland is rich in blood vessels and located on the dorsal and posterior walls of the third ventricle, releasing melatonin into the cerebrospinal fluid and blood at night [15]. Melatonin binds mainly to albumin in the blood, and is metabolized to 6-hydoroxymelatonin by cytochrome P450 (*CYP1A2*). Then, melatonin is conjugated to 6-sulfatoxymelatonin in the liver and excreted in the urine [18]. Melatonin is degraded into N-acetyl-N2-inyl-5-methoxykynuramine (*AFMK*) in the CNS, and *AFMK* is deformylated with N-acetyl-5-methosykynuramine (*AMK*) [19].

### 2.2. Action Mechanism of Melatonin

Melatonin can act either by interacting with specific receptors or directly in the absence of such interactions. Melatonin is a very powerful natural antioxidant that can directly activate the intracellular antioxidant system, as well as chelating oxygen and nitrogen reactive species [20]. Melatonin also acts through melatonin receptors. There are two types of melatonin receptors in mammals—*MT1* and *MT2* [21]. These receptors are heterotrimeric Gi/Go and Gq/11 protein-coupled receptors, interacting with adenylyl cyclase, phospholipase A2, and phospholipase C, reducing *cAMP* and *cGMP* production and activating diacylglycerol and IP3 formation [22]. *MT1* and *MT2* receptors are present in almost all peripheral tissues, as well as in the CNS [23]. Similar to other hormones, melatonin acts as a direct and immediate molecular effector. The rapid effect of melatonin, which affects antioxidant activity and cell signaling, differs depending on the target tissue [24].

## 3. The Role of Melatonin in the CNS

### 3.1. Source of Melatonin in the Human Body

Foods such as nuts, seeds, and vegetables contain melatonin [25]. The intake of melatonin-containing foods increases plasma melatonin levels [26], which are maintained at very low levels during the day and increase to over 50 pg/mL at night. However, plasma melatonin levels are lower than melatonin levels in certain organs in the body (e.g., bone marrow). Therefore, plasma melatonin do not represent concentrations of all organs in the body [26].

### 3.2. Melatonin in the CNS

Previously, both melatonin in the blood and CNS were considered to originate only in the pineal gland. However, Stefulj et al. showed that the *mRNA* of *AANAT* and *HIOMT* is expressed in most regions of the rat brain [27]. According to Liu et al., melatonin can be synthesized in cultured rat cortical astrocytes [28]. Therefore, melatonin may be produced and secreted in areas of the brain other than the pineal gland. As previously mentioned, melatonin is converted to a metabolite, such as *AFMK* or *AMK*, and *AFMK* is the main metabolite in the CNS [29]. Recently, *AFMK* and *AMK* were shown to remove reactive oxygen by forming 3-indolinone, cinnoline, and quinazoline compounds. Melatonin reacts with various reactive intermediates to protect against tissue damage [30]. MT1 receptors are widely distributed in the nervous system, including the hippocampus, caudate putamen, suprachiasmatic nucleus (SCN), a number of novel sites, including the paraventricular nucleus (PVN), periventricular nucleus, supraoptic nucleus (SON), sexually dimorphic nucleus, diagonal band of Broca, nucleus basalis of Meynert, infundibular nucleus, ventromedial and dorsomedial nucleus, tuberomamillary nucleus, mamillary body, and paraventricular thalamic nucleus [31,32]. Unlike the *MT1* receptor, *MT2* receptors are mostly observed in the *CA3* subfield of the hippocampus, reticular thalamic nucleus, supraoptic nucleus, inferior colliculus, substantia nigra pars reticulata, and ventrolateral periaqueductal gray [33]. *MT1* and *MT2* receptors are also distributed in neurons and glial cells of the cerebral cortex, cerebellar cortex, thalamus, and pineal gland [33,34]. Jilg et al. found that the expression of a clock protein, which regulates the circadian rhythm in mice, was regulated by the MT1 melatonin receptor [35]. Looking at the melatonin function described so far, the role of a free radical scavenger can be considered as the main mechanism of action. However, considering that the concentration of melatonin in the human body is lower than that of free radicals, it should be considered that melatonin may have an indirect antioxidant effect through the receptor mechanism, rather than direct antioxidant effects [36,37]. 

## 4. Anti-Ischemic and Antioxidant Effects of Melatonin

Free radical and oxidative stress plays an important role in aging and the development of cancer, atherosclerosis, neurodegenerative diseases, diabetes, and inflammatory diseases [38]. However, the antioxidants created to date have not been proven as therapeutic agents, and new candidate antioxidant drugs are under development. Melatonin acts as an endogenous free radical scavenger and, extensively, as an antioxidant [39]. When melatonin is administered, the expression of antioxidant enzymes, such as superoxide dismutase and glutathione peroxidase, is increased [40]. Therefore, melatonin-induced neuroprotective action is presumably induced by the antioxidative action of melatonin. Upon stroke onset, complex cell damage occurs, resulting in increased cytotoxicity, ROS production, and inflammatory response. An active energy metabolism is essential for the survival of neurons. Reduction in cerebral blood flow, such as in ischemic stroke, and the resulting decrease in glucose and oxygen supply, are fatal to neurons [41]. In the absence of glucose and oxygen, cell survival is affected by various mechanisms. In particular, the deterioration of Na+ K+-ATPase function begins [41,42]. Due to the reduced Na+/K+-ATPase activity, the function of the ATP-dependent pump is reduced and intracellular Na+ is accumulated. Thus, anoxic depolarization is promoted, resulting in activation of the voltage-gated calcium channels and reduction of the Na2+-Ca2+ exchange. Consequently, the Ca2+ accumulates intracellularly, and the cell injury process is initiated [43]. Melatonin is a prominent antioxidant and known to prevent ischemic injury [44]. In studies using animal models of stroke, melatonin was shown to have neuroprotective effects [45]. Administration of melatonin in animal models of stroke was shown to reduce the area of cerebral infarction [46,47]. Melatonin was shown to exert protective effects in both gray and white matter, as well as decrease inflammatory responses and permeability of the blood–brain barrier [48,49,50]. According to research by Letechipía-Vallejo et al., neurologic deficit scores in ischemic vehicle-treated cats were significantly higher than in ischemic melatonin-treated cats seven days after the ischemic episode [51]. Intracerebral transplantation of the pineal gland promoted the secretion of melatonin, which improved motor deficits and symptoms of stroke and reduced the size of the infarct area, based on a rat model of middle cerebral artery (MCA) occlusion [52]. In a study with a rat model of subarachnoid hemorrhage, melatonin injection prevented oxidative brain injury [53]. Melatonin plays a role in Ca2+ homeostasis, and it inhibited the increase of Ca2+ induced by acidification and prevented the increase of Ca2+ caused by glutamate in the cerebral cortex of rats [54,55]. In addition, melatonin can protect mice against stroke by activating *MT2* receptors, which reduces oxidative/inflammatory stress [56]. Li et al. reported that the inhibition of *Nox2* and *Nox4* expressions by melatonin may essentially contribute to its antioxidant and anti-apoptotic effects during brain ischemia-reperfusion injury [57]. Bhattacharya et al. also reported that melatonin at 60 min post-ischemia exerted neuroprotection, as evidenced by reduction in cerebral infarct volume, improvement in motor and neurological deficits, and reduction in brain edema. Furthermore, the ischemia-induced increase in nitrite and malondialdehyde (*MDA*) levels were found to be significantly reduced in ischemic brain regions in treated animals. Melatonin potentiated intrinsic antioxidant status, inhibited acid-mediated rise in intracellular calcium levels, decreased apoptotic cell death, and markedly inhibited protein kinase C (*PKC*)-influenced *AQP4* expression in the cerebral cortex and dorsal striatum [54]. Studies have also been conducted regarding antioxidant effects of melatonin on human subjects. Aly et al. conducted a prospective trial of melatonin on 45 newborns—30 with hypoxic–ischemic encephalopathy (HIE), and 15 healthy controls. The authors found the melatonin/hypothermia group had fewer seizures on follow-up EEG, and less white matter abnormalities on MRI. At 6 months, the melatonin/hypothermia group had improved survival without neurological or developmental abnormalities. The authors concluded that early administration of melatonin to asphyxiated term neonates is feasible and may ameliorate brain injury [58]. Based on another study with newborn babies, significant reductions were observed in malondialdehyde and nitrite/nitrate levels at both 12 and 24 h in the asphyxiated newborns who were given melatonin. Three of the 10 asphyxiated newborns not given melatonin died within 72 h after birth; however, none of the 10 asphyxiated newborns given melatonin died. The results indicated that melatonin may be beneficial in the treatment of newborn infants with asphyxia. The protective actions of melatonin in that study may be associated with the antioxidant properties of indole, as well as the ability of melatonin to increase the efficiency of mitochondrial electron transport [59]. In studies regarding the effects of melatonin on inflammatory responses, intravenous administration of melatonin upon reperfusion effectively decreased the migration of circulatory neutrophils and macrophages/monocytes into the injured brain of rats and inhibited focal microglial activation following cerebral ischemia-reperfusion [49]. Paredes et al. found a significant time-dependent increase in *IL-1β, TNF-α, BAD*, and *BAX* in the ischemic area of both the hippocampus and cortex in the group without melatonin administration. The authors also observed that melatonin administration could significantly reverse these alterations [60]. The results indicate that melatonin has antioxidant and anti-inflammatory properties and exerts neuroprotective effects (Figure 2 and Figure 3).

## 5. Antioxidant Effects of Melatonin in Alzheimer’s Disease (AD)

Oxidative stress plays an important role in the development of Alzheimer’s disease (AD). The generation of free radicals caused by Aβ deposition, mitochondrial dysfunction, and inflammation is very high in AD patients [61,62]. In particular, Aβ plaques cause oxidative stress, which not only promotes lysosomal production but also disrupts lysosomal membrane function, eventually killing cells [63]. In recent studies, a novel mechanism by which melatonin stimulates nonamyloidogenic processing and inhibits the amyloidogenic processing of the β-amyloid precursor protein (βAPP) by stimulating α-secretases and consequently downregulating both β- and γ-secretases at the transcriptional level has been shown [64]. The long-term melatonin treatment in APP+PS1 double-transgenic mice significantly reduced cortical mRNA expression of three antioxidant enzymes (SOD-1, glutathione peroxidase, and catalase) [65]. Rudnitskaya et al. also observed that melatonin administration prevented the decrease in the mitochondria-occupied portion of the neuronal volume and improved the ultrastructure of mitochondria in the hippocampal neurons of the *CA1* region in *OXYS* rats. *OXYS* rats have a phenotype similar to human geriatric disorders, including accelerated brain aging [66]. In addition, *OXYS* rats treated with melatonin showed a decrease in anxiety and impeded deterioration of reference memory [67]. The authors also reported that melatonin significantly increased hippocampal synaptic density and the number of excitatory synapses, decreased the number of inhibitory synapses, and upregulated synapsin I and PSD-95 proteins. Furthermore, melatonin improved the ultrastructure of neuronal and glial cells and reduced glial density [68]. Clinically, melatonin levels were significantly reduced in AD patients [69]. However, clinical trials on the effects of melatonin in patients with Alzheimer’s disease have been reported differently than in preclinical studies. According to a meta-analysis by Wang et al. in seven studies (*n* = 462) with the duration ranging from 10 days to 24 weeks, AD patients receiving melatonin treatment showed prolonged total sleep time at night. However, melatonin did not improve cognitive abilities assessed by the mini-mental state examination and the Alzheimer’s Disease Assessment Scale-Cognition (ADAS-Cog). The discontinuation rate was similar between the melatonin and placebo groups [70]. Also, based on the research results of Xu et al., melatonin therapy may be effective in improving sleep efficacy and prolonging total sleep time in patients with dementia; however, there is no evidence that this improvement impacts cognitive function [71]. Contrary to these findings, Wade et al. found that the median ADAS-Cog values improved statistically when prolonged-release melatonin was added to the existing treatment for Alzheimer’s disease [72]. 

## 6. Antidepressant-Like Actions of Melatonin 

Reportedly, circadian rhythm and melatonin may be associated with depression [73,74]. Agomelatine, a *MT1* and *MT2* receptor agonist, is the first melatonin receptor ligand to demonstrate antidepressant effects in animal model experiments [75,76]. In a study with Flinders Sensitive Line rats, Overstreet et al. showed that agomelatine reduced immobility time in the forced swim test (FST) [77]. The Flinders Sensitive Line (FSL) rat was originally proposed as an animal model of depression because, like depressed humans, it is supersensitive to the behavioral and hormonal effects of cholinergic (muscarinic) agonists [78]. In another study, treatment with agomelatine completely normalized stress-affected cell survival and partly reversed reduced doublecortin expression in the hippocampi of rats in the chronic mild stress model [79]. Agomelatine can regulate dopamine by blocking 5-HT2 receptors and exhibits antidepressant effects in humans, similar to the antidepressant effects of melatonin [80]. Mantovani et al. reported the immobility time in the tail suspension test (TST) of mice was reduced by intraperitoneal (0.1–30 mg/kg) or intracerebroventricular (0.001–0.1 nmol/site) administration of melatonin. The authors also observed a sub-effective dose of melatonin injected intraperitoneally (0.001 mg/kg) produced a synergistic antidepressant-like effect with MK-801, ketamine, zinc chloride, and imipramine in the TST. The authors speculated that the antidepressant effects of melatonin were due to the interaction of melatonin with *NMDA* receptors and the L-arginine-NO pathway [81]. Based on a study conducted with a chronic mild stress (CMS) model, five weeks of unpredictable chronic mild stress in mice decreased grooming and increased serum corticosterone levels. The unpredictable chronic mild stress-induced changes were counteracted by melatonin and imipramine [82]. The antidepressant effects of melatonin are associated with the activation of melatonin receptors. According to Micale et al. [83], the effects of melatonin on immobility time of rats in the FST paradigm was abolished by the simultaneous injection of the non-selective melatonin antagonist, luzindole. Melatonin may have an antidepressant effect by increasing the expression of *GABAA* receptors in the rat brain. 

Melatonin also increases neuroplasticity in the hippocampus, which plays an important role in the development of depression. *MT1* and *MT2* receptors in the hippocampus are mainly distributed in the dentate gyrus, *CA3*, *CA1*, and subiculum. The activation of *MT1* and *MT2* receptors in the subgranular zone of the dentate gyrus promotes cell proliferation. Reportedly, melatonin stimulates dendrite formation by activating *CaMKII* and *PKC* [84]. Based on another study, the *MT2* receptor activates *Akt/GSK-3β/CRMP*-2 signaling, and is necessary and sufficient to mediate functional axonogenesis and synaptic formation [85]. The results indicate that melatonin can regulate neuroplasticity in the hippocampus, similar to the currently used antidepressants. The preclinical findings indicate the antidepressant effects of melatonin; however, the results from clinical studies in humans have not established the antidepressant effects of melatonin. Although most clinical studies of melatonin have been conducted in patients with sleep disorders, results from a few studies have indicated melatonin may affect mood, including depression and improved metabolic side-effects of atypical antipsychotics [86,87]. Therefore, to prove that melatonin is effective in the treatment of depression, further well-designed and large-scale studies should be conducted.

## 7. Conclusions

In this review, the role of melatonin in nerve cell protection, neuroprotection caused by antioxidant effects of melatonin, and the effects of melatonin on neuroplasticity were summarized. Melatonin not only plays a role as an antioxidant, but also acts as an anti-excitotoxicity and anti-inflammatory molecule in neurons. Furthermore, melatonin is a neuroprotective molecule and potential antidepressant because it can pass through the blood–brain barrier and has few side effects [83]. Therefore, melatonin-acting receptors, particularly *MT1* and *MT2*, could be important therapeutic targets for the development of new drugs, including antidepressants.

## Figures and Tables

**Figure 1 brainsci-09-00285-f001:**
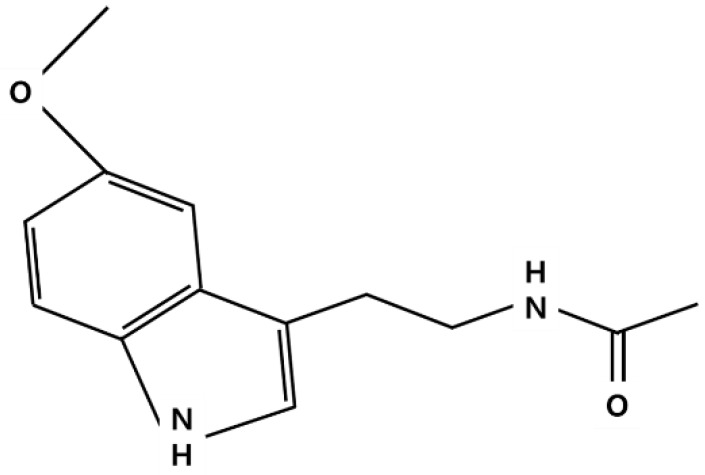
Chemical structure of melatonin.

**Figure 2 brainsci-09-00285-f002:**
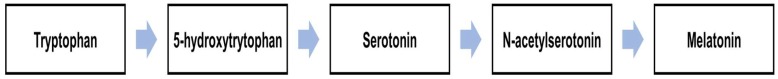
Synthesis of melatonin.

**Figure 3 brainsci-09-00285-f003:**
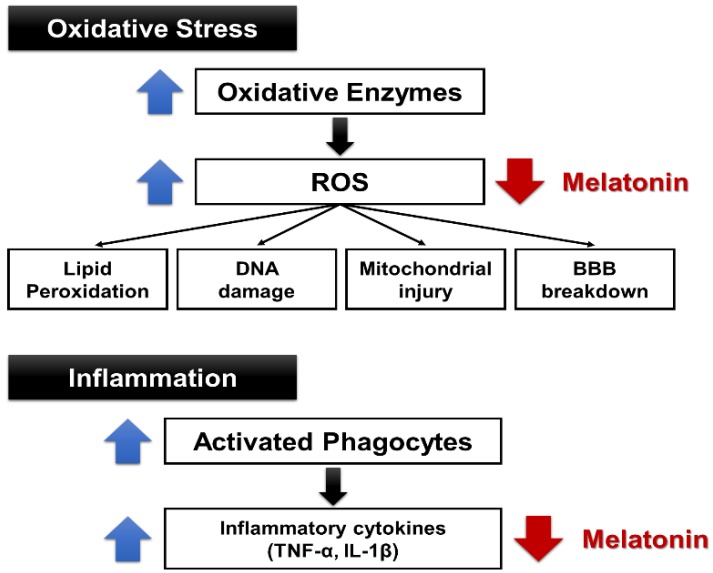
Antioxidant and anti-inflammatory properties of melatonin. BBB: blood–brain barrier.

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
