# Peer review of "The Neuroprotective Effects of Melatonin: Possible Role in the Pathophysiology of Neuropsychiatric Disease"

_brainsci, 2019, doi:10.3390/brainsci9100285_

Round 1

Reviewer 1 Report

The paper by Jung Goo Lee and co-authors shortly analyze the recent scientific evidence of the neuroprotective effects mediated by the hormone melatonin.

Although this perspective does not appear particularly innovative, it could be of potential interest for the readership of “Brain Sciences” because it summarizes some purported protective functions of melatonin in the central nervous system that are recently emerging.

Major points:

In my opinion the first two chapter of the review are rather disorganized with many sentences that are not consequential to what precedes. Several pleonastic sentences are present in the text (see for example, lines 60-63 in which the authors repeat what is written in lines 56-60), so that the authors should reduce redundancy in their text.

The title of chapter 3 (The role of melatonin in human body and CNS) does not reflect the content of chapter 3 because the role of melatonin in haman body is not discussed at all, while it is just mentioned for what concerns the CNS.

Throughout the text the authors describe the potential function of melatonin and its metabolites as free radical scavengers. Nevertheless, such a purported role could be discussed considering the concentrations of these molecules in relation of the concentrations of reactive oxygen species. In my opinion, considering the low concentrations of melatonin inside the body this function is unlikely and an indirect action on the cellular antioxidant response, mediated by its receptor, could be more plausible.

When describing the effects of melatonin on AD (chapter 5), the authors should also discuss whether melatonin has been used in clinical trials and, in case, the results observed in such trials.

Sometimes, the experimental set-up of the works cited by the authors is not clear. More specifically, they should describe more in deep the OXYS rats model as well as Flinders Sensitive Line rats. Which phenotypes do characterize these animal models?

Minor points

In chapter 2.2 I would suggest to add a first sentence to introduce the concept that melatonin can act either by interacting with specific receptors or directly in the absence of such interactions. This would simplify the comprehension of the following text.

Lines 91-92: the sentence “Therefore, plasma melatonin levels do not represent plasma concentrations of all organs…” should be “Therefore, plasma melatonin levels do not represent concentrations of all organs…”.

In the abstract (line 24) melatonin is described as a “natural neurotransmitter-like molecule”. I would rather define melatonin as a hormone.

Lines 130-132: the sentence “Thus, anoxic depolarization is promoted, resulting in activation of the voltage-gated calcium channel and Na+-Ca2+ exchange” should become: “Thus, anoxic depolarization is promoted, resulting in activation of the voltage-gated calcium channels and reduction of Na+-Ca2+ exchange”.

A list of the abbreviations or acronyms used should be added

Lines 205-207: the sentence reporting the results of a specific study should be further discussed in the context of the general discussion about the effects of melatonin in Alzheimer’s disease.

Author Response

We are very grateful to the reviewers for their feedback on our manuscript. All of the suggested corrections have been made.

Reviewer 1

< Major points >

In my opinion the first two chapter of the review are rather disorganized with many sentences that are not consequential to what precedes. Several pleonastic sentences are present in the text (see for example, lines 60-63 in which the authors repeat what is written in lines 56-60), so that the author should reduce redundancy in their text.

We thank you for the good comments. I have made the following corrections to your message.

Then, 5-hydroxytryptophan is converted to serotonin, which is converted to N-acetylserotonin by arylalkylamine N-acetyltransferase (AANAT) and, finally, to melatonin by acetylserotonin O-methyltransferase [14,15]. AANAT and hydroxyindole-O-methyltransferase (HIOMT) are important enzymes involved in the production of melatonin in the central nervous system (CNS). AANAT converts serotonin to N-acetylserotoin, which in turn converts N-acetylserotoin to melatonin [3].

The title of chapter 3 (The role of melatonin in human body and CNS) does not reflect the contents of chapter 3 because the role of melatonin in haman body is not discussed at all, while it is just mentioned for what concerns the CNS.

We thank you for the good comments. Corrected the title you used to point out..

3. The role of melatonin in the CNS

3.1. Source of melatonin in the human body

Foods such as nuts, seeds and vegetables contain melatonin [25]. The intake of melatonin-containing foods increases plasma melatonin levels [26], which are maintained at very low levels during the day and increase to over 50pg/mL at night. However, plasma melatonin levels are lower than melatonin levels in certain organs in the body (e.g., bone marrow). Therefore, plasma melatonin levels do not represent plasma concentrations of all organs in the body [26].

Throughout the text the authors describe the potential function of melatonin and its metabolites as free radical scavengers. Nevertheless, such a purported role could be discussed considering the concentrations of these molecules in relation of the concentrations of reactive oxygen species. In my opinion, considering the low concentrations of melatonin inside the body this function is unlikely and an indirect action on the cellular antioxidant response, mediated by its receptor, could be more plausible.

Thank you for making a good point. We have added your comments to section 3.2 and added references. Thank you.

Looking at the melatonin function described so far, the role of free radical scavenger can be considered as the main mechanism of action. However, considering that the concentration of melatonin in the human body is lower than that of free radicals, it should be considered that melatonin may have an antioxidant effect indirectly through the receptor mechanism rather than direct antioxidant effects [36,37].

36.  Xia, M.Z.; Liang, Y.L.; Wang, H.; Chen, X.; Huang, Y.Y.; Zhang, Z.H.; Chen, Y.H.; Zhang, C.; Zhao, M.; Xu, D.X.; Song, L.H. Melatonin modulates TLR4-mediated inflammatory genes through MyD88- and TRIF-dependent signaling pathways in lipopolysaccharide-stimulated RAW264.7 cells. J. Pineal Res. 2012, 53, 325-334.

37.  Overstreet, D.H. The Flinders sensitive line rats: a genetic animal model of depression. Neurosci. Biobehav. Rev. 1993, 17, 51-68.

When describing the effects of melatonin on AD (chapter 5), the authors should also discuss whether melatonin has been used in clinical trials and, in case, the results observed in such trials.

Thank you for a good comment. We added he clinical effects of melatonin on AD to manuscript and added new references.

However, clinical trials on the effects of melatonin in patients with Alzheimer's disease have been reported differently than in preclinical studies. According to meta-analysis of Wang et al. In seven studies (n= 462) with the duration ranging from 10 days to 24 weeks. AD patients receiving melatonin treatment showed prolonged total sleep time at night. However, melatonin did not improve cognitive abilities assessed by the mini-mental state examination and the Alzheimer's Disease Assessment Scale-Cognition (ADAS-Cog). The discontinuation rate was similar between the melatonin and placebo groups [70]. Also, based on the research results of Xu et al. melatonin therapy may be effective in improving sleep efficacy and prolonging total sleep time in patients with dementia; however, there is no evidence that this improvement impacts cognitive function [71]. Contrary to these findings, Wade et al. found that median the ADAS-Cog values improved statistically when prolonged-release melatonin was added to the existing Alzheimer's disease treatment [73].

70.  Wang, Y.Y.; Zheng, W.; Ng, C.H.; Ungvari, G.S.; Wei, W.; Xiang, Y.T. Meta-analysis of randomized, double-blind, placebo-controlled trials of melatonin in Alzheimer's disease. Int. J. Geriatr. Psychiatry 2017, 32, 50-57.

71.  Xu, J.; Wang, L.L.; Dammer, E.B.; Li, C.B.; Xu, G.; Chen, S.D.; Wang, G. Melatonin for sleep disorders and cognition in dementia: a meta-analysis of randomized controlled trials. Am. J. Alzheimers Dis. Other Demen. 2015, 30. 439-447.

72.  Wade, A.G.; Farmer, M.; Harari, G.; Fund, N.; Laudon, M.; Nir, T.; Frydman-Marom, A.; Zisapel, N. Add-on prolonged-release melatonin for cognitive function and sleep in mild to moderate Alzheimer's disease: a 6-month, randomized, placebo-controlled, multicenter trial. Clin. Interv. Aging 2014, 9, 947-961.

Sometimes, the experimental set-up of the works cited by the authors is not clear. More specifically, they should describe more in deep the OXYS rats model as well as Flinders Sensitive Line rats. Which phenotypes do characterize these animal models?

Thank you for the good comment. Added descriptions for OXYS rats and Flinders sensitive rats.

Rudnitskaya et al. also observed that melatonin administration prevented the decrease in the mitochondria-occupied portion of the neuronal volume and improved the ultrastructure of mitochondria in the hippocampal neurons of the CA1 region in OXYS rats. OXYS rats have a phenotype similar to human geriatric disorders, including accelerated brain aging [66].

66.  Fan, R.; Schrott, L.M.; Arnold, T.; Snelling, S.; Rao, M.; Graham, D.; Cornelius, A.; Korneeva, N.L. Chronic oxycodone induces axonal degeneration in rat brain. BMC Neurosci. 2018, 19, 15

Overstreet et al. showed in a study with Flinders Sensitive Line rats that agomelatine reduced immobility time in the forced swim test (FST) [77]. The Flinders Sensitive Line (FSL) rat was originally proposed as an animal model of depression because, like depressed humans, it is supersensitive to the behavioral and hormonal effects of cholinergic (muscarinic) agonists [78].

78.  Overstreet, D.H. The Flinders sensitive line rats: a genetic animal model of depression. Neurosci. Biobehav. Rev. 1993, 17, 51-68.

< Minor points >

In chapter 2.2 I would suggest to add a first sentence to introduce the concept that melatonin can act either by interacting with specific receptors or directly in the absence of such interactions. This would simplify the comprehension of the following text.

Thank you for the good comment. I corrected the contents as you said.

Melatonin can act either by interacting with specific receptors or directly in the absence of such interactions.

Lines 91-92: the sentence “Therefore, plasma melatonin levels do not represent plasma concentrations of all organs…” should be Therefore, plasma melatonin levels do not represent concentrations of all organs…”.

Thank you for the good comment. I corrected the contents as you said.

Therefore, plasma melatonin do not represent concentrations of all organs in the body.

In the abstract (line 24) melatonin is described as a natural neurotransmitter-like molecule”. I would rather define melatonin as a hormone.

Thank you for the good comment. I corrected the contents as you said.

Melatonin is a hormone that is secreted by the pineal gland.

Lines 130-132: the sentence “Thus, anoxic depolarization is promoted, resulting in activation the voltage-gated calcium channel and Na+-Ca2+ exchange” should become: “Thus, anoxic depolarization is promoted, resulting in activation of the voltage-gated calcium channels and reduction of Na -Ca exchange”.

Thank you for the good comment. I corrected the contents as you said.

Thus, anoxic depolarization is promoted, resulting in activation of the voltage-gated calcium channels and reduction of Na2+-Ca2+ exchange.

A list of the abbreviations or acronyms used should be added.

Thank you for the good comment. As you said, we add a list of abbreviation.

Lines 205-207: the sentence reporting the results of a specific study should be further discussed in the context of the general discussion about the effects of melatonin in Alzheimer’s disease.

Thank you for a good comment. We added he clinical effects of melatonin on AD to manuscript and added new references.

However, clinical trials on the effects of melatonin in patients with Alzheimer's disease have been reported differently than in preclinical studies. According to meta-analysis of Wang et al. In seven studies (n= 462) with the duration ranging from 10 days to 24 weeks. AD patients receiving melatonin treatment showed prolonged total sleep time at night. However, melatonin did not improve cognitive abilities assessed by the mini-mental state examination and the Alzheimer's Disease Assessment Scale-Cognition (ADAS-Cog). The discontinuation rate was similar between the melatonin and placebo groups [70]. Also, based on the research results of Xu et al. melatonin therapy may be effective in improving sleep efficacy and prolonging total sleep time in patients with dementia; however, there is no evidence that this improvement impacts cognitive function [71]. Contrary to these findings, Wade et al. found that median the ADAS-Cog values improved statistically when prolonged-release melatonin was added to the existing Alzheimer's disease treatment [73].

Reviewer 2 Report

The review by Lee et al., “The neuroprotective effects of melatonin: possible role in the pathophysiology of neuropsychiatric disease” attempts to focus on the role of melatonin in nerve cell protection or neuroprotection caused by antioxidant effects of melatonin. The topic addressed is interesting, and useful contribution. The authors, however, should remove redundancies of some content (section 4. Anti-ischemic and antioxidant effects of melatonin) so that it is understandable to the readers. On the other hand, abbreviations, e.g., Nox, OXYS, and AQP4 should be explained more carefully.

Author Response

We are very grateful to the reviewers for their feedback on our manuscript. All of the suggested corrections have been made.

Reviewer 2

The review by Lee et al., “The neuroprotective effects of melatonin: possible role in the pathophysiology of neuropsychiatric disease” attempts to focus on the role of melatonin in nerve cell protection or neuroprotection caused by antioxidant effects of melatonin. The topic addressed is interesting, and useful contribution. The authors, however, should remove redundancies of some content (section 4. Anti-ischemic and antioxidant effects of melatonin) so that it is understandable to the readers. On the other hand, abbreviations, e.g., Nox, OXYS, and AQP4 should be explained more carefully.

à Thanks for the good comments. As you said, I have reduced the content of section 4 and made a list of abbreviations.